# Preferences and uptake of home-based HIV self-testing for maternal retesting in Kenya

Alison L. Drake[1,2]*, Wenwen Jiang[2], Peninah Kitao[3], Shiza Farid[1], Barbra A. Richardson[1,4], David A. Katz[1], Anjuli D. Wagner[1], Cheryl C. Johnson[5], Daniel Matemo[3], GraceJohn Stewart[1,2,6,7], John Kinuthia[3]

1 Department of Global Health, University of Washington, Seattle, WA, United States of America, 2 Department of Epidemiology, University of Washington, Seattle, WA, United States of America, 3 Research and Programmes, Kenyatta National Hospital, Nairobi, Kenya, 4 Department of Biostatistics, University of Washington, Seattle, WA, United States of America, 5 Department of Global HIV, Hepatitis and STI Programmes, World Health Organization, Geneva, Switzerland, 6 Department of Medicine, University of Washington, Seattle, WA, United States of America, 7 Department of Pediatrics, University of Washington, Seattle, WA, United States of America

* adrake2@uw.edu

## Abstract

### Objective

To compare preferences, uptake, and cofactors for unassisted home-based oral self-testing (HB-HIVST) versus clinic-based rapid diagnostic blood tests (CB-RDT) for maternal HIV retesting.

### Design

Prospective cohort.

### Methods

Between November 2017 and June 2019, HIV-negative pregnant Kenyan women receiving antenatal care were enrolled and given a choice to retest with HB-HIVST or CB-RDT. Women were asked to retest between 36 weeks gestation and 1-week post-delivery if the last HIV test was <24 weeks gestation or at 6 weeks postpartum if ≥24 weeks gestation, and self-report on retesting at a 14-week postpartum.

### Results

Overall, 994 women enrolled and 33% (n = 330) selected HB-HIVST. HB-HIVST was selected because it was private (n = 224, 68%), convenient (n = 211, 63%), and offered flexibility in the timing of retesting (n = 207, 63%), whereas CB-RDT was selected due to the trust of providers to administer the test (n = 510, 77%) and convenience of clinic testing (n = 423, 64%). Among 905 women who reported retesting at follow-up, 135 (15%) used HB-HIVST. Most (n = 595, 94%) who selected CB-RDT retested with this strategy, compared to 39% (n = 120) who selected HB-HIVST retesting with HB-HIVST. HB-HIVST retesting was more common among women with higher household income and those who may have been unable to test during pregnancy (both retested postpartum and delivered <37 weeks

**Data Availability Statement:** We have included the link to the data at figshare at the following DOI: 10.6084/m9.figshare.26129440.

**Funding:** a University of Washington (UW) Center for AIDS Research (CFAR) New Investigator Award, supported by National Institutes of Health/National Institute of Allergy and Infectious Diseases P30-AI027757 to ALD and K01-AI116298 to ALD. https://www.niaid.nih.gov/ The sponsors/funders did not play any role in the study design, data collection and analysis, decision to publish, or preparation of the manuscript.

**Competing interests:** The authors have declared that no competing interests exist.

gestation) and less common among women who were depressed. Most women said they would retest in the future using the test selected at enrollment (99% [n = 133] HB-HIVST; 93% [n = 715] CB-RDT-RDT).

## Conclusions

While most women preferred CB-RDT for maternal retesting, HB-HIVST was acceptable and feasible and could be used to expand HIV retesting options.

## Introduction

Eliminating pediatric HIV by 2030 is a global priority, and maternal retesting and linkage to care have been highlighted by the Global Alliance to help curb new infections due to high maternal HIV incidence and elevated risk of vertical HIV transmission associated with incident HIV infection [1–3]. Retesting at specific time points offers a programmatic mechanism to test again later and capture prior delayed or missed antenatal care (ANC) visits, or as a result of limited staffing or test kit stock-outs. WHO issued maternal HIV retesting recommendations during pregnancy, with catch-up testing at delivery or 6 weeks gestation, and an additional postpartum retest for those with ongoing HIV risk in high HIV burden settings [4]. Many countries are striving to achieve elimination, and have adopted policies recommending maternal retesting [5]. In Kenya, retesting is recommended in the third trimester, delivery, at 6 weeks and 6 months postpartum, and every 6 months thereafter while breastfeeding [6, 7]. However, implementation of WHO guidance on maternal retesting has been challenging, possibly due to limited resources, policy complexity, or lack of perceived benefit.

Implementation of retesting during pregnancy is variable, and retesting data beyond delivery is scarce. In Kenya, retesting was significantly higher postpartum than in pregnancy/delivery [8]. In Zambia, while retesting in pregnancy was universal among women who returned, total coverage was only 67% due to missed visits [9]. In South Africa, a similar proportion (64%) of women eligible for retesting were retested during pregnancy, while delivery retesting was uncommon (17%) [10]. In other sub-Saharan African countries, 25–40% of eligible women retested during pregnancy [11–14].

Many studies have shown HIV self-testing (HIVST) increases HIV testing uptake [4, 15], including studies that asked pregnant/postpartum women to encourage their male partners to test, which can help women select prevention interventions, such as pre-exposure prophylaxis (PrEP) [16–20]. This testing approach is safe, highly acceptable, and easy to use, and self-testers can reliably conduct and interpret results [21–23]. HIVST for maternal retesting may help fill programmatic gaps, as women have prior testing experience in pregnancy, and may be more comfortable self-testing. Among pregnant and postpartum Kenyan women offered HIVST, 54% preferred clinic-based HIVST to standard blood testing [24]. Clinic-based HIVST may overcome retesting barriers, such as fear of blood collection and wait times for providers to test [25]. Home-based HIVST may overcome additional barriers, such as duration of clinic visits, comfort with setting, and testing at a time when women are ready to test. WHO updated HIVST recommendations, stating this testing modality should be offered as a testing approach, which could help maintain essential HIV services during times when services are disrupted, such as during the COVID-19 pandemic, as well as at facilities to support PrEP [15, 26, 27]. While guidelines highlight the utility of offering multiple HIVST service delivery models, specific guidance on how they can be used to facilitate maternal retesting has not been provided [15].

We offered HIV-negative pregnant women the option of retesting with clinic-based rapid diagnostic tests (CB-RDT) using blood samples or home-based oral self-tests (HB-HIVST) to measure preferences for, and uptake and cofactors of, HB-HIVST. Results from this study may help policymakers assess HB-HIVST as a potentially viable option for maternal retesting and their partners, as a complementary testing strategy to identify incident HIV infections.

## Materials and methods

### Study design and population

We conducted a prospective cohort study between November 2017 and June 2019 Kenya in Nairobi, Kenya (Riruta Health Center [urban]) and Western Kenya (Ahero County Hospital and Bondo sub-County Hospital [rural]); these high HIV burden areas have an antenatal HIV prevalence of 19% and 16%, respectively [28]. Women seeking ANC at maternal and child health (MCH) clinics were screened for study eligibility. Women who were pregnant (age ≥14 years, age 14–17 emancipated minors), had documentation of a prior HIV-negative test during pregnancy, were willing to retest for HIV, and had daily access to a mobile phone were eligible. Written informed consent was obtained before study participation. All study procedures were approved by the Kenyatta National Hospital/University of Nairobi Ethics and Research Committee (#P788/11/2016) and the University of Washington Human Subjects Division (#STUDY00000414).

### Data collection

At enrollment, study nurses administered a survey on a tablet using Open Data Kit, including demographic and clinical characteristics, partner characteristics, HIV risk factors, and perceptions of HIVST at home and using blood samples at the clinic. Data on HIV testing history, gestational age, and syphilis status were abstracted from the mother's MCH booklet. Depression was measured on the Edinburgh Postnatal Depression Scale (EPDS) [29] and relationship power using the Sexual Relationship Power Scale (SRPS) [30]. Maternal retesting was scheduled based on the timing of the last HIV test and by national Kenyan guidelines, which recommend retesting in the third trimester of pregnancy, delivery, and at 6 weeks and 6 months postpartum [6]. The timing of testing was scheduled to avoid retesting <3 months after their last test to align with guidelines. If the last HIV test during pregnancy was <24 weeks gestation, women were asked to retest between 36 weeks gestation and 1-week post-delivery; if ≥24 weeks gestation, they were asked to retest at 6 weeks postpartum.

Women were given a choice retesting strategy for the first retest following an initial HIV-negative test in pregnancy, either standard of care testing in clinics (CB-RDT) or HIVST to conduct at home (HB-HIVST) unassisted. Standard of care tests in Kenya included Alere Determine HIV-1/2 test [Abbott Laboratories, Abbott Park, IL] and First Response HIV-1-2-0 [Premier Medical Corporation Ltd., Kachigam, India], with the Uni-Gold Recombigen HIV-1/2 (Trinity Biotech, Wicklow, Ireland) to confirm diagnosis. The OraQuick ADVANCE Rapid HIV-1/2 Antibody Test (OraSure Technologies, Inc., Bethlehem, PA) was used by the research team with HIVST instructions for use in Kenya for the study using graphical illustrations and accompanying Kiswahili and English text to avoid literacy restrictions. This assay is WHO prequalified and has high sensitivity (99.3%) and specificity (99.8%) [31]. Study staff counseled women on retesting options and asked a series of questions to help women make informed decisions, including their ability and comfort with getting to the clinic, comfort with blood vs. oral fluid samples, and plans for confirmatory testing and support if tests were reactive. They were also asked about HIV self-test kit storage, where testing could occur, and comfort with test conduct and interpretation.

Pregnant women who selected HB-HIVST were also offered the opportunity to take up to three HIV self-test kits home for male partner testing. Those who selected CB-RDT were told they could refer their partner to the clinic for individual or couples-based testing. Additional questions to assess comfort with offering male partners HB-HIVST included comfort asking the partner to test, perceptions of partners' testing preferences, and comfort performing the partner's HIV self-test if requested. Women were also asked about plans for partner confirmatory testing if results were reactive and if there were any safety concerns (i.e., partner violence) if women used a self-test or asked their partner to self-test at home. Those who selected HB-HIVST were initially given HIV self-tests for themselves and their partner (if applicable) to take home after enrollment; the protocol was later modified to allow participants to return and pick up self-test kits at a later date to avoid longer-term storage. While women were asked to choose either CB-RDT or HB-HIVST initially, women could change their minds at any point in the study.

Participants were asked for permission for nurses to contact them via phone call or short message service (SMS) one week before their retest date. Participants who selected CB-RDT at enrollment were tested when they returned to the clinic by study nurses using the standard of care national algorithm. Participants were asked to 'flash' (call and then hang-up) study nurses after they completed HB-HIVST; nurses then called participants during business hours to inquire about the test outcome or gave further instructions if women had difficulty conducting the test or reported invalid results. Post-test counseling regarding prevention of mother-to-child HIV transmission (PMTCT) guidelines was provided to all participants, including the need to come to the clinic for confirmatory testing if they obtained reactive results. HIV retest type and results were recorded at the time of testing for women retested with CB-RDT, and as soon as they were reported for women with HB-HIVST. Follow-up visits scheduled at 14 weeks postpartum retrospectively assessed retesting outcomes, and nurses administered surveys to women (either in-person or via phone) to inquire about their and their partner's test experiences, preferences, and linkage to further testing, prevention, and treatment. CB-RDT results were observed and recorded by study staff while HB-HIVST and all male partner HIV testing were self-reported by women.

## Statistical analysis

We aimed to enroll 1000 women to detect a 10% difference in testing modes, assuming 25% would select HB-HIVST, $\alpha = 0.05$, 90% power, and two-sided testing; we enrolled 997 of 1029 women screened. Women were classified as having at least minor depression if the EPDS score was >10. Low relationship power was defined as an overall score in the lowest quantile from the study population on the SRPS. Preterm birth (PTB) was defined as a <37 weeks gestation at delivery. Chi-squared tests were used to compare categorical variables. Generalized linear models (GLM) with Poisson-link and robust standard errors were constructed to identify characteristics of women interested and able to retest with HB-HIVST; separate models were constructed for selecting HB-HIVST at enrollment and retesting with HB-HIVST by 14 weeks postpartum (vs. CB-RDT) and restricting the latter analysis to only women who completed retesting. Study site and age were identified as *a priori* potential confounders in both models and included in the multivariable models along with variables with p-values<0.10 in univariate models. An interaction term between PTB and timing of completion of retesting in the completion of the retesting model was included since some women who intended to retest with HB-HIVST may not have been able to do so if they delivered early. An exploratory analysis to characterize factors associated with completing retesting using HB-HIVST among women who selected HB-HIVST as their testing strategy at enrollment was also conducted.

For all models, sensitivity analyses were conducted among women who completed study follow-up <3 months of the estimated 14-week postpartum date to assess potential bias among women with late follow-up visits. Data were analyzed using RStudio Version 1.2.5042 (RStudio, Inc, Boston, MA).

## Results

Among 1,029 pregnant women screened for eligibility, 997 (97%) were eligible, of which 994 (99%) were enrolled (Fig 1). The median age was 24 (interquartile range [IQR] 21–27) years (Table 1). Most (n =, 84%) women were married or cohabitating, 6% (n = 54) were in polygamous relationships, and 13% (n = 123) had been in their current relationship for <1 year. Over one-quarter of women with partners did not know their partner's HIV status. Half (n = 547) of the women had at least minor depression. Most (n = 696, 70%) used transportation to get to the clinic, with 21% (n = 208) reporting >1 hour in travel time. Clinic waiting time was >1 hour for 34% (n = 334) of women, and 13% (n = 125) reported leaving the clinic due to long wait times in the past. Some (n = 123, 12%) women said the clinic hours did not work with their daily schedule.

### HIV retest selected at enrollment

The majority (n = 330, 67%) selected CB-RDT for retesting; most said they selected this option because they trusted providers to administer the test (n = 510, 77%) and thought it was convenient to test at the clinic (n = 423, 64%). Familiarity and reliability with blood tests were also cited as reasons for selecting CB-RDT. Of the one-third of women who selected HB-HIVST, most said they selected this option because self-testing was private (n = 224, 68%) and convenient (n = 211, 63%) (Fig 2A); oral sample collection would be easy to do (n = 177, 54%) and

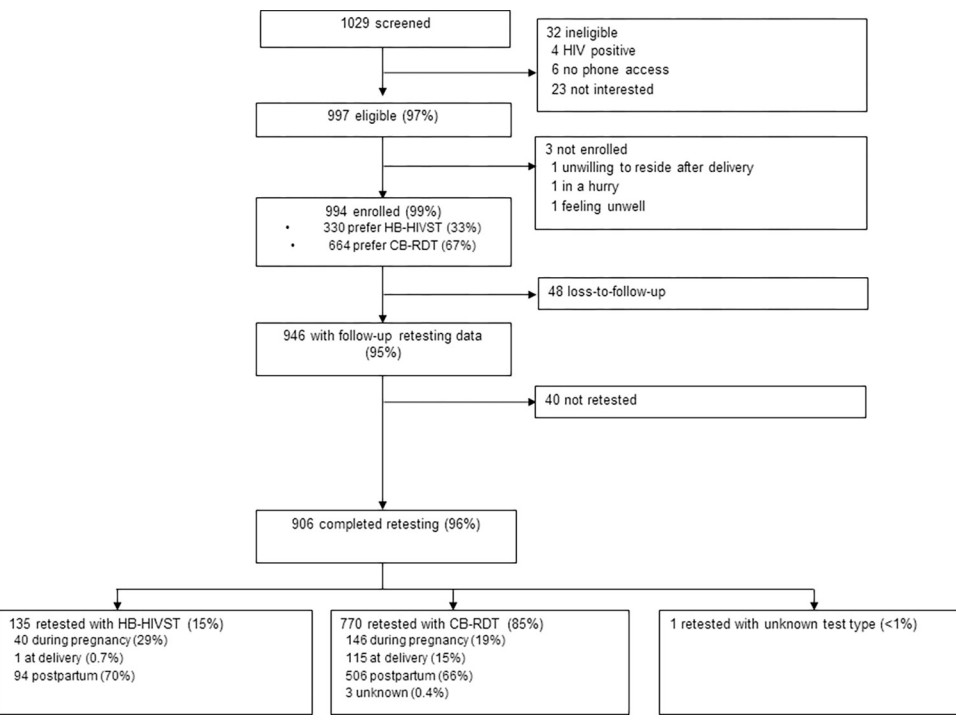

**Fig 1. Study flowchart.** Home-based HIV self-test (HB-HIVST); clinic-based rapid diagnostic test (CB-RDT).

**Table 1. Characteristics of study participants at enrollment (N = 994).**

| | N | n (%) or median (IQR) |
|---|---|---|
| *Sociodemographic characteristics* | | |
| Study site | 994 | |
| Nairobi | | 460 (46) |
| Western Kenya | | 534 (54) |
| Age (years) | 994 | 24 (21–27) |
| Gestational age at enrollment (week) | 994 | 28.0 (22.0–32.0) |
| Gestational age ≥24 weeks at enrollment | 994 | 724 (73) |
| Completed secondary education | 994 | 510 (51) |
| Employed | 994 | 335 (34) |
| Household income ≥10,000 (KSH) per month | 876 | 318 (36) |
| Ever diagnosed with STI | 988 | 27 (3) |
| Depression[a] | 994 | 547 (55) |
| Number of living children | 994 | 1 (0–2) |
| Current pregnancy intended | 994 | 582 (59) |
| *Partner characteristics* | | |
| Marital status | 994 | |
| Married or cohabitating | | 832 (84) |
| Not married, with partner | | 67 (7) |
| No partner | | 95 (10) |
| Current relationship polygamous (vs. monogamous)[b] | 925 | 54 (6) |
| Current relationship duration (years) [b] | 926 | |
| <1 | | 123 (13) |
| 1–5 | | 473 (51) |
| >5 | | 330 (36) |
| Low partnership power[c] | 895 | 226 (25) |
| Partner age difference (years) | 889 | 5 (3–7) |
| Partner HIV status | 927 | |
| Positive | | 8 (1) |
| Negative | | 675 (73) |
| Unknown | | 244 (26) |
| *Clinic experiences* | | |
| Travel time to clinic ≥1 hour[d] | 993 | 208 (21) |
| Used transportation to clinic[d] | 992 | 696 (70) |
| Clinic wait time ≥1 hour[d] | 993 | 334 (34) |
| Ever left clinic because of long wait | 993 | 125 (13) |
| Clinic hours inconvenient | 994 | 123 (12) |

Interquartile range (IQR); Kenyan Shilling (KSH), sexually transmitted infection (STI); home-based HIV self-test (HB-HIVST); clinic-based rapid diagnostic test (CB-RDT). KSH ~ $1 USD. a. score >10 on Edinburgh Postnatal Depression Scale (EDPS); b. among women who reported having a partner; c. score in lowest quantile (<2.15) on Sexual Relationship Power Scale (SRPS); d. last clinic visit before enrollment

offered flexibility to test when women wanted to test (n = 207, 63%) (Fig 2B); and 59% (n = 196) said testing at home was more convenient (Fig 2C).

Selection of HB-HIVST (vs CB-RDT) was more common among women who completed secondary education and reported higher household income. Poorer health system factors, including traveling ≥1 hour to the clinic (25% vs 19%, respectively; p = 0.03), waiting ≥1 hour at the clinic (39% vs. 31%, respectively; p = 0.01), and inconvenient clinic hours (17% vs. 10%,

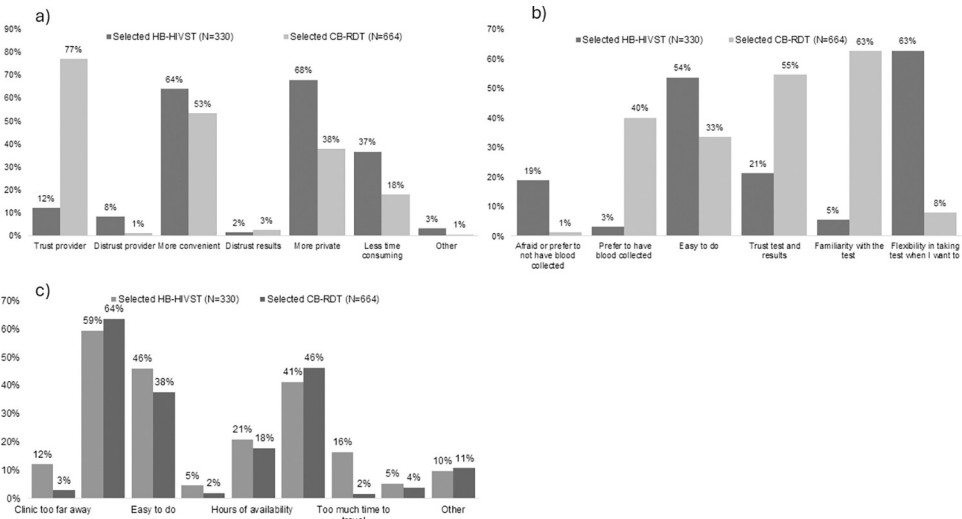

**Fig 2. Preferences for HIV retesting at enrollment, by method selected (n = 994).** Home-based HIV self-test (HB-HIVST); clinic-based rapid diagnostic test (CB-RDT). a. Reasons cited for preference of self or provider administered, b. Reasons cited for preference of oral fluid or blood test, c. Reasons cited for preference of home or clinic.

respectively; p<0.01) were also associated with selecting HB-HIVST. In contrast, HB-HIVST selection was less common among women who had at least mild depression. The proportion of women who were ≥24 weeks gestation at enrollment was similar between those who selected HB-HIVST (71%) and CB-RDT (74%) (p = 0.4, S1 Fig). Among 925 women with partners, women in polygamous relationships were less likely to select HB-HIVST (3% vs. 7%, respectively; p = 0.01). After adjusting for study site and age, completing secondary education (adjusted prevalence ratio [aPR] 1.38, 95% CI: 1.04–1.85; p = 0.03), higher household income (aPR 1.43, 95% CI: 1.02–2.01; p = 0.04), and inconvenient clinic hours (aPR 2.26, 95% CI: 1.44–3.57; p<0.01) remained significantly associated with HB-HIVST.

## Completion of HIV retesting

Among 946 (95%) women with follow-up data, 906 (96%) retested: 769 (81%) with CB-RDT, 135 (15%) with HB-HIVST, and 1 (<1%) with an unknown type. The overall attrition rate was 9.2% Only 1 woman was detected as having an incident HIV infection, with a positive CB-RDT during the follow-up period Most (n = 601, 67%) women retested postpartum (>48 hours post-delivery), 186 (20%) during pregnancy, and 116 (14%) ≤48 hours of delivery. Among 444 women scheduled to retest by delivery with retesting follow-up data, 418 (94%) retested; 179 during pregnancy/delivery (43%), and 237 (53%) postpartum. Among 663 women scheduled to retest during pregnancy who reported retesting, 27% (n = 180) delivered preterm. Among 237 women with postpartum retests, 51% (n = 121) delivered preterm. Among 808 women who retested and reported partner testing, half (n = 363, 46%) had partners who also retested.

Among 946 women with follow-up data, most (n = 595, 94%) of the 635 women who selected CB-RDT retested with CB-RDT; in contrast, only 39% (n = 120) of 311 who selected HB-HIVST retested with HB-HIVST while 56% (n = 175) retested with CB-RDT (Fig 3A). Overall, 15% (n = 135) of 905 women retested with HB-HIVST. A similar proportion of women who selected HB-HIVST vs. CB-RDT were not retested (5% vs. 4%, p = 0.5). Among 295 women who selected HB-HIVST, the ability to complete retesting with HB-HIVST was

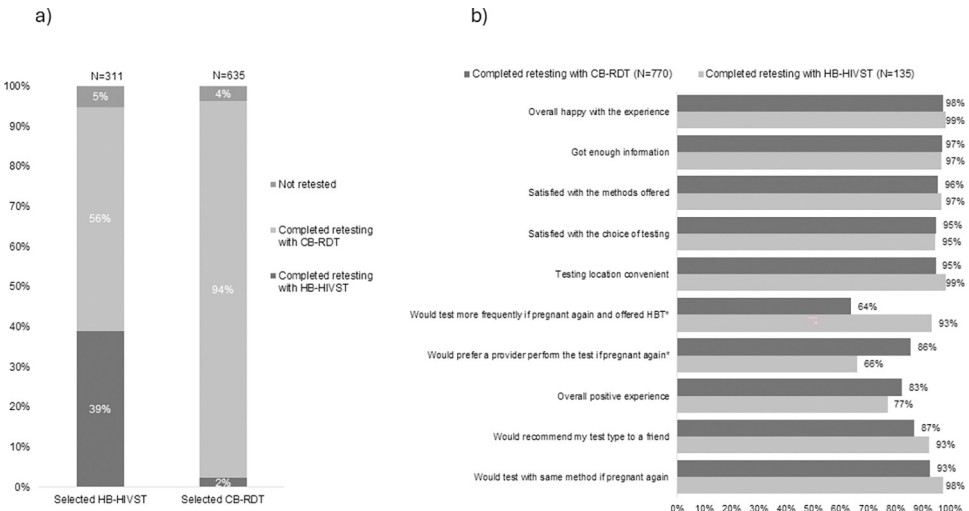

**Fig 3. HIV retesting status and future test preference among women with follow-up, by testing option selected at enrollment.** Fig 3B questions were answered based on self-reported responses of "Strongly Agree" or "Agree". * p<0.05 by Chi-square test. Home-based HIV self-test (HB-HIVST); clinic-based rapid diagnostic test (CB-RDT). a. HIV retesting status among women with follow-up visits, by testing modality selected at enrollment (N = 946), b. Future test preferences, by test type used (N = 905).

less likely among women who lived in Western Kenya, delivered preterm, and was more likely among women with higher household income and partners who tested (S2 Table). In a sensitivity analysis among 261 women who selected HB-HIVST and completed follow-up within 3 months of the estimated 14-week postpartum date, depression was the only significant co-factor of HB-HIVST retesting with a similar effect size as the primary analysis (PR 0.71, 95% CI: 0.53–0.96; p = 0.02) (S3 Table).

Among 905 women who retested, women who used HB-HIVST (vs CB-RDT) were less likely to be from Western Kenya (37% vs. 53%, respectively; p<0.01) and have least mild depression (37% vs. 58%, respectively; p<0.01) (Table 2). In contrast, women who retested with HB-HIVST were more likely to deliver preterm (77% vs. 68% [term]; p = 0.03), be married/cohabiting (90% vs. 80% [not married/cohabitating]; p = 0.04), have a household income ≥10,000 KSH (52% vs. 35% [<10,000]; p<0.01), wait ≥1 hour at the clinic (42% vs. 32% [<1 hour]; p = 0.02), and have a partner tested for HIV during follow-up (74% vs. 41% [partner not tested]; p<0.01). After adjusting for study site and age, depression (aPR 0.40, 95%CI: 0.25–0.67; p<0.01), higher household income (aPR 1.62, 95%CI: 1.04–2.51; p = 0.03) and having a partner tested (aPR 5.67, 95%CI: 3.34–9.61; p<0.01) remained significantly associated with HB-HIVST retesting. In addition, women who retested postpartum after a PTB were significantly less likely to use HB-HIVST than women who tested postpartum after delivering at term. Results were similar in sensitivity analyses among women with follow-up data collected within 3 months of the 14-week postpartum follow-up visit (S4 Table).

## Future retesting preference

The majority (n = 127, 93%) of the 135 women who retested with HB-HIVST said they would test more often if they became pregnant again and were offered HB-HIVST, while 66% (n = 90) said they would prefer a provider to test (Fig 3B). Most women indicated they would be willing to use the same retesting strategy again, with higher proportions of women who used HB-HIVST than CB-RDT reporting a future preference for the same test type (98% vs.

**Table 2. Correlates of completing HIV retesting with HB-HIVST by 14 weeks postpartum (vs. CB-RDT) (N = 905).**

| | Retested with HB-HIVST (N = 135) | | Retested with CB-RDT (N = 770) | | | | | |
|---|---|---|---|---|---|---|---|---|
| | N | n (%) or median (IQR) | N | n (%) or median (IQR) | Crude PR (95% CI) | p | Adjusted PR (95% CI) [d] | p |
| Western Kenya (ref: Nairobi) | 135 | 50 (37) | 770 | 406 (53) | 0.58 (0.41–0.81) | <0.01* | 0.94 (0.54–1.65) | 0.84 |
| Age (year) | 135 | 24 (22, 28) | 770 | 24 (21, 27) | 1.02 (0.99–1.05) | 0.24 | 1.00 (0.96–1.05) | 0.85 |
| Gestational age ≥24 weeks at enrollment | 135 | 97 (72) | 770 | 567 (74) | 0.93 (0.67–1.28) | 0.64 | | |
| Preterm birth (gestational age at delivery<37 weeks) | 135 | 104 (77) | 769 | 520 (68) | 0.66 (0.46–0.95) | 0.03* | | |
| Tested in pregnancy | | | | | | | 1.23 (0.55–2.75) | 0.62 |
| Tested in postpartum | | | | | | | 0.32 (0.17–0.59) | <0.01* |
| Completed secondary education | 135 | 76 (56) | 770 | 390 (51) | 1.21 (0.89–1.66) | 0.23 | | |
| Employed | 135 | 48 (36) | 770 | 260 (34) | 1.07 (0.78–1.47) | 0.68 | | |
| Household income ≥10,000 (KSH) per month | 119 | 62 (52) | 687 | 240 (35) | 1.82 (1.30–2.53) | <0.01* | 1.62 (1.04–2.51) | 0.03* |
| Depression[a] | 135 | 50 (37) | 770 | 450 (58) | 0.48 (0.34–0.66) | <0.01* | 0.40 (0.25–0.67) | <0.01* |
| Have live births | 135 | 79 (59) | 770 | 434 (56) | 1.08 (0.80–1.46) | 0.63 | | |
| Current pregnancy intended | 135 | 81 (60) | 767 | 451 (59) | 1.04 (0.77–1.42) | 0.79 | | |
| Married/cohabiting[b] | 135 | 122 (90) | 770 | 640 (83) | 1.76 (1.03–3.02) | 0.04* | 1.52 (0.58–3.98) | 0.40 |
| Relationship duration <1 year[b] | 129 | 11 (9) | 718 | 93 (13) | 0.67 (0.37–1.20) | 0.17 | | |
| Low partnership power[c] | 129 | 34 (26) | 718 | 179 (25) | 1.07 (0.74–1.53) | 0.73 | | |
| Ever tested with STI | 134 | 2 (1) | 766 | 23 (3) | 0.53 (0.14–2.04) | 0.36 | | |
| Traveling time to clinic ≥1 hour[d] | 135 | 29 (21) | 769 | 162 (21) | 1.02 (0.70–1.49) | 0.91 | | |
| Using transportation to clinic[d] | 134 | 83 (62) | 769 | 541 (70) | 0.73 (0.53–1.00) | 0.05 | 0.90 (0.59–1.39) | 0.64 |
| Waiting time ≥1 hour at clinic[d] | 135 | 57 (42) | 770 | 247 (32) | 1.43 (1.05–1.96) | 0.02* | 1.41 (0.92–2.15) | 0.12 |
| Ever left clinic because of long wait | 135 | 23 (17) | 769 | 89 (12) | 1.45 (0.97–2.17) | 0.07 | 1.58 (0.87–2.87) | 0.13 |
| Schedule not working with clinic hours | 135 | 20 (15) | 770 | 80 (10) | 1.40 (0.92–2.14) | 0.12 | | |
| Tested during postpartum (ref: pregnancy/delivery) | 135 | 94 (70) | 767 | 506 (66) | 1.15 (0.83–1.61) | 0.40 | ** | ** |
| Partner tested for HIV during follow-up[e] | 130 | 96 (74) | 678 | 279 (41) | 3.26 (2.28–4.66) | <0.01* | 5.67 (3.34–9.61) | <0.01* |

Prevalence ratio (PR); Confidence interval (CI); home-based oral test (HB-HIVST); clinic-based blood test (CB-RDT)

[a] assessed by Edinburgh Postnatal Depression Scale (EDPS) with a score of >10

[b] married / cohabiting (vs. no partner)

[c] score in lowest quantile (<2.15) on Sexual Relationship Power Scale (SRPS)

[d] assessed with the last clinic visit before enrollment

[e] among women with partners with HIV-negative or unknown status who reported partner testing status during follow-up; reparametrized in the multivariate model to allow for women without partners to be included in the model.

* p<0.05

** Includes as interaction term with preterm birth. Kenya Shilling (KSH) ~ $1 USD.

93%, respectively; p<0.0001). Willingness to use the same retesting strategy was also similar among women who used HB-HIVST and CB-RDT when women who completed follow-up >3 months after their scheduled 14-week postpartum date were excluded from the analysis (S1 Fig).

## Discussion

HB-HIVST was highly acceptable for maternal retesting, with one-third selecting this testing strategy over CB-RDT. This proportion is similar to the proportion of women who reported willingness to self-test during pregnancy/delivery in Nigeria [32]. Retesting coverage was

higher for CB-RDT (85%) than HB-HIVST (15%), despite nearly one-third of women selecting HB-HIVST at enrollment. These results demonstrate a current preference and better uptake for CB-RDT, but also barriers to HB-HIVST among those who selected this approach. Women who lived in Western Kenya, had at least mild depression, delivered preterm, and had lower household income were less likely to complete HB-HIVST suggesting there may be both logistical and psychosocial challenges to self-testing. While not noted in our study, it is also possible that the need for confirmatory testing, if self-tests are reactive and/or not providing an option to test at facilities, could have deterred women from selecting this testing strategy. However, HB-HIVST could still contribute to overcoming gaps in retesting coverage by addressing barriers to retesting with CB-RDT previously reported, including constrained provider time and the need for confidential space for testing [25]. As familiarity and experience with HIVST increases in Kenya, demand for this testing approach is expected to increase over time.

Reasons for selecting HB-HIVST included privacy, flexibility, and ease of use. In contrast, trust and familiarity with blood testing with providers, and trust of test and results were reasons for preferring HB-HIVST. These results concur with findings from a prior study in Kenya which found the primary reasons pregnant women selected HIVST in a clinic setting were lack of pain with blood draws, privacy, ease of use, and procedure timeliness [24]. Women in our study were more likely to select HB-HIVST if they found clinic hours were inconvenient, which may suggest that convenience may be important in testing preferences. However, logistical barriers to conducting HB-HIVST may have led to a lower uptake of this approach as the clinic schedule was not associated with HB-HIVST retesting. In addition, women with at least mild depression were less likely to select and use HB-HIVST, which may indicate a lack of motivation required to learn and utilize this testing option. Together these findings suggest heterogeneity in underlying reasons for testing preferences, but a high willingness to retest with the test that best suits their needs.

Our study had several strengths. While the proportion of women who retested during pregnancy/ delivery (34%) was lower than the 47% we anticipated based on gestational age at enrollment, early postpartum testing would most likely capture HIV infections acquired in pregnancy due to lower sexual activity early in the postpartum period [6]. In addition, one-quarter of women who retested were scheduled to retest during pregnancy delivered preterm. Distribution of HIVST kits for women at high risk of PTB could facilitate higher coverage of retesting. While retesting late (i.e, postpartum) is likely partially attributed to early delivery, other logistical or behavioral factors may also explain late retesting. Therefore, flexibility in timing of retesting reflects real-life "catch-up" testing approaches that include retesting at delivery or 6-week postpartum visits and is supported by both Kenyan national guidelines and WHO [4, 7]. Recent modeling results suggest that "catch-up" testing is the most cost-effective approach in Kenyan PMTCT programs [33]. We utilized the choice of test type to mimic real-world settings where HB-HIVST could be used as a complementary strategy for maternal retesting, rather than assigning a strategy women would prefer not to use. We captured retesting preferences both at enrollment when women selected their testing modality and after retesting, which captures intentions before and after their retesting experiences. Women were also provided an opportunity to test with their partner if they selected HB-HIVST, or refer partners for CB-RDT; this approach supports partner engagement with a direct link to testing, and other studies have found secondary distribution strategies for male partner testing can improve male partner test coverage [16, 17]. However, HB-HIVST may be challenging for women who are uncomfortable asking partners to test. Therefore, alternative strategies that support women in different types of partnerships to encourage partner testing are necessary.

Our findings are also subject to limitations. Results may not be generalizable to other settings where HB-HIVST is more common, in more rural settings, or outside of Kenya. Timing and setting of delivery may have led to higher CB-RDT if women delivered in facilities and were offered CB-RDT, or if providers offered CB-RDT before the study visit; however, providers at study sites were supportive of retesting through the study. Timing of follow-up visits was variable; 10% of women had visits >3 months late. In sensitivity analyses, differences in future retesting preferences by test type used were no longer significant when follow-up visits >3 months late were excluded. Since male partner testing can co-occur with maternal retesting, we included male partner testing as a co-factor for retesting strategies; however, maternal retesting may also be an exposure for male partner testing. Finally, we did not capture the timing of male partner testing relative to maternal retesting, which limits our ability to understand the desire for testing together and use of HB-HIVST as a testing strategy.

Maternal HIV retesting will increasingly help curb vertical HIV transmission and help close the gaps towards the UNAIDS 2025 targets of 95% of pregnant and breastfeeding women living with HIV receiving testing and 95% having suppressed viral loads. Offering women choices to meet HIV retesting needs, respecting privacy, confidentiality, convenience, and trust of individuals can help achieve these targets. Expanding HIV testing options may also address gaps in service delivery, such as disruptions caused by COVID-19 or healthcare worker strikes, and increasing access to HIVST has potential to provide more consistent availability of HIV testing in the future.

## Supporting information

**S1 Table. Correlates of selecting HB-HIVST as a retesting strategy at enrollment (vs. CB-RDT) (N = 994).**
(DOCX)

**S2 Table. Correlates of completing HIV retesting with HB-HIVST (vs. CB-RDT) among women who selected HB-HIVST at enrollment (N = 295) [#].**
(DOCX)

**S3 Table. Correlates of completing HIV retesting with HB-HIVST by 14 weeks postpartum (vs. CB-RDT) among women who selected HB-HIVST and completed follow-up within 3 months of the estimated 14-week postpartum date (N = 251).**
(DOCX)

**S4 Table. Correlates of completing HIV retesting with HB-HIVST (vs. CB-RDT) by 14 weeks postpartum among women who completed retesting with a follow-up visit within 3 months of the estimated 14 weeks postpartum date (N = 813).**
(DOCX)

**S1 Fig. Future test preference among women who completed follow-up within 3 months of the estimated 14-week postpartum date, by retesting type at follow-up (N = 813).** Questions were answered based on self-reported responses of "Strongly Agree" or "Agree". * $p < 0.05$ by Chi-square test. Home-based HIV self-test (HB-HIVST); clinic-based rapid diagnostic test (CB-RDT).
(TIF)

## Author Contributions

**Conceptualization:** Alison L. Drake, David A. Katz, Daniel Matemo, GraceJohn Stewart, John Kinuthia.

**Data curation:** Alison L. Drake, Peninah Kitao, Shiza Farid, Daniel Matemo, John Kinuthia.

**Formal analysis:** Alison L. Drake, Wenwen Jiang, Barbra A. Richardson, Anjuli D. Wagner.

**Funding acquisition:** Alison L. Drake, Barbra A. Richardson, David A. Katz, GraceJohn Stewart, John Kinuthia.

**Methodology:** Alison L. Drake, Peninah Kitao, Barbra A. Richardson, Anjuli D. Wagner, Daniel Matemo, GraceJohn Stewart.

**Project administration:** Alison L. Drake, Peninah Kitao, Shiza Farid.

**Supervision:** Alison L. Drake, Peninah Kitao, John Kinuthia.

**Visualization:** Alison L. Drake, Wenwen Jiang.

**Writing – original draft:** Alison L. Drake, Wenwen Jiang.

**Writing – review & editing:** Alison L. Drake, Wenwen Jiang, Peninah Kitao, Shiza Farid, Barbra A. Richardson, David A. Katz, Anjuli D. Wagner, Cheryl C. Johnson, Daniel Matemo, GraceJohn Stewart, John Kinuthia.

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
