## [Decision Letter · Decision Letter 0]

15 May 2024

PONE-D-24-11702Preferences and uptake of home-based HIV self-testing for maternal retesting in KenyaPLOS ONE

Dear Dr. Drake,

Thank you for submitting your manuscript to PLOS ONE. After careful consideration, we feel that it has merit but does not fully meet PLOS ONE’s publication criteria as it currently stands. Therefore, we invite you to submit a revised version of the manuscript that addresses the points raised during the review process.

Please submit your revised manuscript by Jun 29 2024 11:59PM. If you will need more time than this to complete your revisions, please reply to this message or contact the journal office at plosone@plos.org. Please include the following items when submitting your revised manuscript:A rebuttal letter that responds to each point raised by the academic editor and reviewer(s). You should upload this letter as a separate file labeled 'Response to Reviewers'.A marked-up copy of your manuscript that highlights changes made to the original version. You should upload this as a separate file labeled 'Revised Manuscript with Track Changes'.An unmarked version of your revised paper without tracked changes. You should upload this as a separate file labeled 'Manuscript'.

We look forward to receiving your revised manuscript.

Kind regards,

Hamufare Dumisani Dumisani Mugauri, Ph.D. Public Health

Academic Editor

PLOS ONE

Journal Requirements:

Reviewers' comments:

Reviewer's Responses to Questions

**Comments to the Author**

1. Is the manuscript technically sound, and do the data support the conclusions?

Reviewer #1: Yes

Reviewer #2: Yes

2. Has the statistical analysis been performed appropriately and rigorously? 

Reviewer #1: Yes

Reviewer #2: Yes

3. Have the authors made all data underlying the findings in their manuscript fully available?

Reviewer #1: Yes

Reviewer #2: Yes

4. Is the manuscript presented in an intelligible fashion and written in standard English?

Reviewer #1: Yes

Reviewer #2: Yes

5. Review Comments to the Author

Reviewer #1: Reviewer: Bernard Njau

Title “Preferences and Uptake of Home-based HIV Self-testing for Maternal Retesting in Kenya".

Manuscript ID: PONE-D-24-11702

General Comment:

This research is relevant in the field of HIV/AIDS care and treatment among pregnant women attending antenatal clinics in Kenya. The study is well thought out, designed, and implemented; however, the authors need to address the following comments for improvement:

Abstract:

Results;

i). The authors should present the sample(n=) before presenting the percentages (%) or vice versa. This should be done in the result section as well.

Conclusion:

i) The authors should be more specific in their conclusions based on their study objectives and key findings, and provide specific recommendations for their conclusions. I suggest the authors to recast their conclusion.

Main Text:

Background.

i) In the last paragraph of the background section the authors should add what are the expected benefits of the study findings for pregnant women, their partners, and other secondary beneficiaries.

Methods

i) The authors have used MCH…is it not Reproductive and Child Health(RCH)clinics? See line 24 on page 4.

Ethical consideration;

ii) Authors should provide ethical clearance certificate identification numbers and date/month/year. See lines 2 to 4 on page 5.

Data collection;

i) The authors should describe the source of the questionnaire use and the number of questions (n=? open-ended; n=? close-ended?).

ii) Number of study nurses? Were they trained? How long? On what? What was their previous experience in such research?

Measurement;

i) Authors should add the measurement sub-section and provide sample questions and expected responses for all categories of independent variables and dependent variables.

ii) Measurement for internal variability?

iii) Was piloting testing of the data collection tools done? Among how many samples? What were the aims of doing the piloting and how were the results utilized?

iv) The authors mentioned “saliva”…I’m not sure if it’s correct to mention saliva as the sample used in Oral HIVST…I suggest authors to use the correct word. See line 4 on page 6.

v) The authors mentioned” …. with test conduct and interpretation…” What method/strategy did the authors use to ascertain this observation? See lines 6 & 7 on page 6.

vi) The authors mentioned, “Participants were asked to ‘flash’…”. What about those who did not flash? For how long (hours? days? Weeks?) did the nurses wait for the flash before making a follow-up? How frequently did the nurses attempt to make follow-ups before stopping? See lines 22-23 and lines 6 -9 on page 7.

vii) In the last paragraph under statistical analysis, I suggest the authors add a sentence “We used adjusted Prevalence ratio with their corresponding 95% Confidence Interval to summarize the strength of the association between the independent variables and the dependent variable(s).

Results:

i) Authors should refer to my early comment in the abstract section.

ii) Authors should give the reason for not enrolling the 3 eligible participants.

iii) What was the attrition rate?

iv) Authors should present their findings under the following sub-sections, a) Descriptive findings, b) Bivariate analysis findings, and c) multivariable analysis findings for a reader to follow.

v) The authors mentioned: “…completing secondary education…, and inconvenient clinic hours…...” However, these two observation were not reported in the abstract section. See lines 13 -15 on page 9.

Discussion:

i) The authors mentioned that” HB-HIVST was highly acceptable…” What was the reference of acceptability to justify this statement? See line 19 on page 11.

ii) The following sentence” These findings demonstrate a current preference and better uptake for CB-RDT, but also barriers to HB-HIVST” not convincing. To my understanding, HIVST helps to circumvent barriers associated with facilities-based HTC. I suggest the authors should provide a plausible explanation for the CB-RDT preference in this study setting. See the lines 23-24 on page 11.

iii) The authors mentioned,”…other logistical or behavioral factors…”. I suggest the authors provide examples of those factors. See line 2 on page 13.

Study limitations:

i). The authors should report methodological study limitations, and steps used to minimize the limitations (e.g., self-reporting of HIVST results? Attrition rate? study design? etc.).

Reviewer #2: The article address an important problem in a very specific setting.

To better understand the policy and difficulties for HIV retesting in context, authors should add a short description of antenatal and birth care (i.e. frequency of antenatal care visits and uptake, frequency of delivery in facilities vs at home). Also, please clarify if CB-RDT is performed always for deliveries in facilities.

If retesting is recommended in the third trimester, delivery, at 6 weeks and 6 months postpartum, and every 6 months thereafter while breastfeeding why the study protocol proposed less often retesting?

Authors mentioned that timing of testing was scheduled to avoid retesting <3 months after their last test. What is the reason to avoid more frequent retest? Is this due to cost?

If the last HIV test during pregnancy was <24 weeks gestation, women were asked to retest between 36 weeks gestation and 1 week post-delivery; if ≥24 weeks gestation, they were asked to retest at 6 weeks postpartum.

With these 2 last recommendations, are they not missing the possibility of testing before delivery?

The manuscript does not mention the HIV results for the HIV tests performed. Is this information available? Does it change by testing strategy? If no results available, the decision to ignore test results should be explained and also discussed as a limitation.

6. PLOS authors have the option to publish the peer review history of their article (what does this mean?). If published, this will include your full peer review and any attached files.

Reviewer #1: No

Reviewer #2: No

---

## [Author Response · Author response to Decision Letter 0]

1 Jul 2024

Reviewer #1: Reviewer: Bernard Njau

Title “Preferences and Uptake of Home-based HIV Self-testing for Maternal Retesting in Kenya".

Manuscript ID: PONE-D-24-11702

General Comment:

This research is relevant in the field of HIV/AIDS care and treatment among pregnant women attending antenatal clinics in Kenya. The study is well thought out, designed, and implemented; however, the authors need to address the following comments for improvement:

Abstract:

Results;

i). The authors should present the sample(n=) before presenting the percentages (%) or vice versa. This should be done in the result section as well.

We have included this consistently through the abstract and the results, with the exception of the model where the emphasis is on the % and the measure of association rather than the N.

Conclusion:

i) The authors should be more specific in their conclusions based on their study objectives and key findings, and provide specific recommendations for their conclusions. I suggest the authors to recast their conclusion.

We removed “and partner testing” from the conclusion as the abstract results did not present any partner data. We believe the conclusions on acceptability and feasibility match the objective of measuring uptake, and are appropriate to the results; however we did modify the conclusion with a more direct recommendation appropriate to our findings.

Main Text:

Background.

i) In the last paragraph of the background section the authors should add what are the expected benefits of the study findings for pregnant women, their partners, and other secondary beneficiaries.

We added potential benefits to the end of the background section.

Methods

i) The authors have used MCH…is it not Reproductive and Child Health(RCH)clinics? See line 24 on page 4.

At these facilities they are referred to as MCH, we have spelled out the acronym here as well.

Ethical consideration;

ii) Authors should provide ethical clearance certificate identification numbers and date/month/year. See lines 2 to 4 on page 5.

We have added the approval numbers as per PLoS guidelines but not the dates as they span multiple years and is not required by PLoS. We have confirmed with PLoS that approval was prior to study initiation.

Data collection;

i) The authors should describe the source of the questionnaire use and the number of questions (n=? open-ended; n=? close-ended?).

We respectfully disagree that this level of detail is necessary for this study as the questionnaire was not the central focus of the study and is not required or standard in PLoS to include this level of detail.

ii) Number of study nurses? Were they trained? How long? On what? What was their previous experience in such research?

We respectfully disagree that this level of detail is necessary for this study. It also is not clear if the reviewer was interested in their clinical training or for this research project. This level of detail is not typically included in PLoS journals of similar types, so we opted not to include it for brevity.

Measurement;

i) Authors should add the measurement sub-section and provide sample questions and expected responses for all categories of independent variables and dependent variables.

We respectfully disagree that his level of detail is necessary for this study. This is not a standardized survey, and This level of detail is not typically included in PLoS journals of similar types, so we opted not to include it for brevity.

ii) Measurement for internal variability?

We did not do this so have not added any additional details.

iii) Was piloting testing of the data collection tools done? Among how many samples? What were the aims of doing the piloting and how were the results utilized?

We pilot tested this informally with the study team but not in a systematic way that we felt warranted description in the manuscript. As this is not a study to validate a survey we do not think this level of detail is required and have opted to omit these details for brevity.

iv) The authors mentioned “saliva”…I’m not sure if it’s correct to mention saliva as the sample used in Oral HIVST…I suggest authors to use the correct word. See line 4 on page 6. 

We have replaced the term “saliva” with “oral fluid”.

v) The authors mentioned” …. with test conduct and interpretation…” What method/strategy did the authors use to ascertain this observation? See lines 6 & 7 on page 6.

The prior sentence indicated that study staff counseled women. This is an extension of counseling to help think through decision making. By stating “They were also...” We feel this is clear this is still part of the counseling process.

vi) The authors mentioned, “Participants were asked to ‘flash’…”. What about those who did not flash? For how long (hours? days? Weeks?) did the nurses wait for the flash before making a follow-up? How frequently did the nurses attempt to make follow-ups before stopping? See lines 22-23 and lines 6 -9 on page 7.

All women were also assessed at 14 weeks postpartum to retrospectively assess this information. We did not have separate protocols for women who flashed and did not flash, nor were there time limits on when women could flash. Nurses did not make repeated attempts to ask women to flash. There was a window period around the 14-week time point for in-person visits; however if women happened to come back after all attempts to reach women were exhausted we allowed data capture for up to 1 year if they happened to come back to the clinic when the study was ongoing. This level of detail is more in line with a standard operating procedure rather than a manuscript, therefore we did not elaborate on these details.

vii) In the last paragraph under statistical analysis, I suggest the authors add a sentence “We used adjusted Prevalence ratio with their corresponding 95% Confidence Interval to summarize the strength of the association between the independent variables and the dependent variable(s).

We respectfully disagree with the reviewer that this is necessary. It is standard practice to calculate measures of association (in this case prevalence ratios) with 95% confidence intervals in multivariable models and we do not think it is necessary to include this level of detail in the interest of brevity.

Results:

i) Authors should refer to my early comment in the abstract section.

We assume the reviewer is referring to the N and % comment and have edited this accordingly.

ii) Authors should give the reason for not enrolling the 3 eligible participants.

This information is already included in Figure 1.

iii) What was the attrition rate?

9.2%, this is added in the results.

iv) Authors should present their findings under the following sub-sections, a) Descriptive findings, b) Bivariate analysis findings, and c) multivariable analysis findings for a reader to follow.

We respectfully disagree that this is necessary. It is not a requested format from PLoS nor typical in most manuscripts to do this. We feel the subheadings by scientific topic area are more informative for the reader and have maintained these.

v) The authors mentioned: “…completing secondary education…, and inconvenient clinic hours…...” However, these two observation were not reported in the abstract section. See lines 13 -15 on page 9.

Yes, we agree with the reviewer. There was not sufficient space to include all of the results in the abstract so we had to pick and choose which ones to highlight in the abstract.

Discussion:

i) The authors mentioned that” HB-HIVST was highly acceptable…” What was the reference of acceptability to justify this statement? See line 19 on page 11.

This is an interpretation of the results, which we clarify after the “acceptable” language, stating “one-third selecting this testing strategy over CB-RDT”

ii) The following sentence” These findings demonstrate a current preference and better uptake for CB-RDT, but also barriers to HB-HIVST” not convincing. To my understanding, HIVST helps to circumvent barriers associated with facilities-based HTC. I suggest the authors should provide a plausible explanation for the CB-RDT preference in this study setting. See the lines 23-24 on page 11.

HIVST in our study included both self-testing with an oral swab AND home based testing. There were clearly barriers to HB-HIVST that led to lower uptake as women who initially selected HB-HIVST had a significantly lower uptake than those with CB-RDT. Preference alone is not driving this, there are other factors. HIVST can help alleviate issues associated with facility based HTC, but the facility based setting is not the only factor; there could be other factors that impacted uptake. The subsequent sentence is the explanation for some results that we found could be related to lower uptake. As this was already provided we did not elaborate further. 

iii) The authors mentioned,”…other logistical or behavioral factors…”. I suggest the authors provide examples of those factors. See line 2 on page 13.

The results summarized immediately before this statement are the basis for these results. Please note that we did not state “other” logistical or behavioral factors, the sentence reads, “Women who lived in Western Kenya, had at least mild depression, delivered preterm, and had lower household income were less likely to complete HB-HIVST suggesting there may be both logistical and psychosocial challenges to self-testing.” Depression, lower income are examples of the logistical and psychosocial challenges and are already explained; therefore we did not alter the manuscript based on this comment.

Study limitations:

i). The authors should report methodological study limitations, and steps used to minimize the limitations (e.g., self-reporting of HIVST results? Attrition rate? study design? etc.).

Reviewer #2: The article address an important problem in a very specific setting.

To better understand the policy and difficulties for HIV retesting in context, authors should add a short description of antenatal and birth care (i.e. frequency of antenatal care visits and uptake, frequency of delivery in facilities vs at home). Also, please clarify if CB-RDT is performed always for deliveries in facilities.

This information was not collected at the facility level, ie ANC visit frequency and facility delivery rates; this was facility level data not individual level data. The study did not collect facility level data so we cannot report on this. Delivery testing would fall under retesting, and it is assumed to be CB-RDT but it is not specified in that language, which means we cannot definitively state whether CB-RDT is “performed always”. This would also fall under facility level data which we did not collect. 

If retesting is recommended in the third trimester, delivery, at 6 weeks and 6 months postpartum, and every 6 months thereafter while breastfeeding why the study protocol proposed less often retesting?

This study was to assess the mode of testing, and as stated in the objective, “to measure preferences for, and uptake and cofactors of, HB-HIVST”. We were not assessing alignment with the retesting guidelines. In addition, retesting uptake is variable during the peripartum period. We assessed the most common recommendation for retesting, the first initial retest after the initial ANC testing. We did add an additional sentence to clarify what we hope the results would help us learn. Notably, this is about the test type, not adherence to the frequency recommended. We did however clarify that this is the first retest following the initial negative test in pregnancy in the methods section.

Authors mentioned that timing of testing was scheduled to avoid retesting <3 months after their last test. What is the reason to avoid more frequent retest? Is this due to cost?

This is to align with guidelines and recommendations. The timing is partly due to cost, and healthcare worker burden, but also the stress of having to retest and the window period between HIV acquisition and a positive result. Quarterly testing is a general guidance for retesting. We did edit the manuscript to clarify it was for alignment with guidelines.

If the last HIV test during pregnancy was <24 weeks gestation, women were asked to retest between 36 weeks gestation and 1 week post-delivery; if ≥24 weeks gestation, they were asked to retest at 6 weeks postpartum.

With these 2 last recommendations, are they not missing the possibility of testing before delivery?

The goal is not to test before delivery, it is to not test too frequently. If women test at >=24 weeks gestation their next test would fall within the 3 month window and would be considered too soon. Retesting at 6 weeks postpartum is the visit that most closely aligns with routine care and would serve as the next time point to test; this also aligns with guidelines for both timing of retesting and time between tests. If someone tested HIV-negative at 36 weeks gestation, they would not be advised to test at 38-40 weeks prior to delivery as the test would be too soon. The 24 week cut-point was to avoid testing too frequently, which also has limited value for capturing incident infections as the window is too small.

The manuscript does not mention the HIV results for the HIV tests performed. Is this information available? Does it change by testing strategy? If no results available, the decision to ignore test results should be explained and also discussed as a limitation.

We thank the author for noting this inadvertent omission. There was only 1 person with an HIV-positive result. We have edited the manuscript to clarify this and included the test type.

---

## [Decision Letter · Decision Letter 1]

1 Aug 2024

Preferences and uptake of home-based HIV self-testing for maternal retesting in Kenya

PONE-D-24-11702R1

Dear Dr. Drake,

We’re pleased to inform you that your manuscript has been judged scientifically suitable for publication and will be formally accepted for publication once it meets all outstanding technical requirements.

Kind regards,

Hamufare Dumisani Dumisani Mugauri, Ph.D. Public Health

Academic Editor

PLOS ONE

Additional Editor Comments (optional):

Reviewers' comments:

Reviewer's Responses to Questions

**Comments to the Author**

1. If the authors have adequately addressed your comments raised in a previous round of review and you feel that this manuscript is now acceptable for publication, you may indicate that here to bypass the “Comments to the Author” section, enter your conflict of interest statement in the “Confidential to Editor” section, and submit your "Accept" recommendation.

Reviewer #1: All comments have been addressed

Reviewer #2: All comments have been addressed

2. Is the manuscript technically sound, and do the data support the conclusions?

Reviewer #1: Yes

Reviewer #2: Yes

3. Has the statistical analysis been performed appropriately and rigorously? 

Reviewer #1: Yes

Reviewer #2: Yes

4. Have the authors made all data underlying the findings in their manuscript fully available?

Reviewer #1: Yes

Reviewer #2: Yes

5. Is the manuscript presented in an intelligible fashion and written in standard English?

Reviewer #1: Yes

Reviewer #2: Yes

6. Review Comments to the Author

Reviewer #1: (No Response)

Reviewer #2: The authors addressed most comments and questions.

The authors addressed most comments and questions.

7. PLOS authors have the option to publish the peer review history of their article (what does this mean?). If published, this will include your full peer review and any attached files.

Reviewer #1: No

Reviewer #2: No

---

## [Editor Report · Acceptance letter]

5 Aug 2024

PONE-D-24-11702R1 

PLOS ONE

Dear Dr. Drake, 

I'm pleased to inform you that your manuscript has been deemed suitable for publication in PLOS ONE. Congratulations! Your manuscript is now being handed over to our production team.

Kind regards, 

on behalf of

Mr Hamufare Dumisani Dumisani Mugauri 

Academic Editor

PLOS ONE